



# Applying self-supervised learning for semantic cloud segmentation of all-sky images

Yann Fabel[1], Bijan Nouri[1], Stefan Wilbert[1], Niklas Blum[1], Rudolph Triebel[2,4], Marcel Hasenbalg[1], Pascal Kuhn[6], Luis F. Zarzalejo[5], and Robert Pitz-Paal[3]

[1]German Aerospace Center (DLR), Institute of Solar Research, 04001 Almeria, Spain
[2]German Aerospace Center (DLR), Institute of Robotics and Mechatronics, 82234, Oberpfaffenhofen-Weßling, Germany
[3]German Aerospace Center (DLR), Institute of Solar Research, 51147 Cologne, Germany
[4]Technical University of Munich, Chair of Computer Vision & Artificial Intelligence, 85748 Garching, Germany
[5]CIEMAT Energy Department, Renewable Energy Division, 28040 Madrid, Spain
[6]EnBW Energie Baden-Württemberg AG, 76131 Karlsruhe, Germany

**Correspondence:** Yann Fabel (yann.fabel@gmail.de), Bijan Nouri (bijan.nouri@dlr.de)

**Abstract.**

Semantic segmentation of ground-based all-sky images (ASIs) can provide high-resolution cloud coverage information of distinct cloud types, applicable for meteorology, climatology and solar energy-related applications. Since the shape and appearance of clouds is variable and there is high similarity between cloud types, a clear classification is difficult. Therefore,

most state-of-the-art methods focus on the distinction between cloudy- and cloudfree-pixels, without taking into account the cloud type. On the other hand, cloud classification is typically determined separately on image-level, neglecting the cloud's position and only considering the prevailing cloud type. Deep neural networks have proven to be very effective and robust for segmentation tasks, however they require large training datasets to learn complex visual features. In this work, we present a self-supervised learning approach to exploit much more data than in purely supervised training and thus increase the model's

performance. In the first step, we use about 300,000 ASIs in two different pretext tasks for pretraining. One of them pursues an image reconstruction approach. The other one is based on the *DeepCluster* model, an iterative procedure of clustering and classifying the neural network output. In the second step, our model is fine-tuned on a small labeled dataset of 770 ASIs, of which 616 are used for training and 154 for validation. For each of them, a ground truth mask was created that classifies each pixel into *clear sky*, *low-layer*, *mid-layer* or *high-layer* cloud. To analyze the effectiveness of self-supervised pretraining,

we compare our approach to randomly initialized and pretrained ImageNet weights, using the same training and validation sets. Achieving 85.8% pixel-accuracy on average, our best self-supervised model outperforms the conventional approaches of random (78.3%) and pretrained ImageNet initialization (82.1%). The benefits become even more evident when regarding precision, recall and intersection over union (IoU) on the respective cloud classes, where the improvement is between 5 and 20% points. Furthermore, we compare the performance of our best model on binary segmentation with a clear-sky library

(CSL) from the literature. Our model outperforms the CSL by over 7% points, reaching a pixel-accuracy of 95%.





# 1 Introduction

Clouds constantly cover large fractions of the globe, influencing the amount of shortwave radiation reflected, transmitted and absorbed by the atmosphere. Therefore, they do not only affect momentarily local temperatures but also play a significant role in global warming (Rossow and Zhang, 1995; Rossow and Schiffer, 1999; Stubenrauch et al., 2013). The actual impact on solar irradiation depends on the properties of individual clouds and is still a current field of research. Automatic detection and specification of clouds can thus help to study their effects in more detail and to monitor changes that may be induced by climate change. Moreover, cloud data is essential for weather forecasts and solar energy applications. In so-called nowcasting systems for example, shortwave solar irradiation is forecasted in intra-minute and intra-hour time frames using ground-based observations. These forecasts proved to be beneficial for the operation of solar power plants (Kuhn et al., 2018) and have the potential to optimize power distribution of solar energy over electricity nets (Perez et al., 2016).

Ground-based measurements and satellite measurements are the two possibilities to continuously observe clouds. While satellite images cover larger areas, they are also less detailed. On the other hand, ground-based sky observations using all-sky imagers have limited coverage but they can be obtained in higher temporal and spatial resolution. Therefore, all-sky imagers represent a valuable addition to satellite imaging that is being studied increasingly. In the past various ground-based camera systems were developed for automatic cloud cover observations (Shields et al., 1998; Long et al., 2001; Widener and Long, 2004). Furthermore, off-the-shelf surveillance cameras are frequently utilized (West et al., 2014; Blanc et al., 2017; Nouri et al., 2019b). They all observe the entire hemisphere using fish-eye lenses with a viewing angle of about 180°. Thereby, typical cloud heights and clouds spread over multiple square kilometers can be monitored. In this work we refer to these cameras as all-sky imagers and the corresponding image as ASI.

Usually, there are two tasks most studies distinguish. In cloud detection, the precise location of a cloud within the ASI is sought, whereas cloud specification aims to provide information about the properties of an observed cloud. The former is achieved by segmenting the image into cloudy and cloudless pixels. For the latter, the common approach is to classify visible clouds into specified categories of cloud types. Often, classification is based on the 10 main cloud genera defined by the world meteorological organization (WMO) (Cohn, 2017). Both tasks are challenging for a number of reasons. For instance, a clear distinction between aerosols and clouds is not always possible from image data. Corresponding to Calbó et al. (2017), clouds are defined by the visible amount of water droplets and ice crystals. In contrast, aerosols comprise all liquid and solid particles independent of their visibility. Still, the underlying phenomenon is the same and there is no clear demarcation between one and the other. Moreover, clouds fragments can extend multiple kilometers into their surroundings, mixing with other aerosols in a so-called twilight zone (Koren et al., 2007). This absence of sharp boundaries makes precise segmentation particularly difficult. For classification, high similarities between cloud types and large variations in spatial extent pose another challenge. Furthermore, the appearances of clouds are influenced by atmospheric conditions, illuminations and distortion effects from the fish-eye lenses. Finally, overlapping cloud layers are especially difficult to distinguish. As a result, many datasets intentionally neglect such multi-layer conditions or consider the prevailing cloud type only.





Traditionally, segmentation and classification were addressed separately. Firstly, both tasks are hard to solve individually.
Secondly, the solution approaches are often very distinct. Most segmentation techniques rely on threshold-based methods in color space. One commonly applied measure is the red-blue ratio (Long et al., 2006) or difference (Heinle et al., 2010) within the RGB color space. Other methods include the green channel as well (Kazantzidis et al., 2012), apply adaptive thresholding (Li et al., 2011), or transform the image into HSI color space (Souza-Echer et al., 2006; Jayadevan et al., 2015). Also super-pixel segmentation, graph models and combination of both have already been studied for threshold-based cloud segmentation (Liu et al., 2014, 2015; Shi et al., 2017). The problem with thresholds is that they depend on many factors, such as sun elevation, a pixel's relative position to the sun/horizon, and current atmospheric conditions. To take these factors into account, clear sky libraries (CSLs) were introduced (Chow et al., 2011; Ghonima et al., 2012; Chauvin et al., 2015; Wilbert et al., 2016; Kuhn et al., 2018). CSLs store historical RGB data from clear sky conditions which is used to compute a reference image of the threshold color feature (e.g. red-blue ratio). By considering the difference image from reference and original color features, detection is more robust.

Apart from manually adjusting thresholds of color features, learning-based methods were examined, too. To identify the most relevant color components, clustering and dimensionality reduction techniques were applied (Dev et al., 2014, 2016). There are also studies on supervised learning techniques for classifying pixels in ASIs such as neural networks, support vector machines (SVMs), random forests and bayesian classifiers (Taravat et al., 2014; Cheng and Lin, 2017; Ye et al., 2019). Lately, also deep learning approaches using convolutional neural networks (CNNs) were presented (Dev et al., 2019; Xie et al., 2020; Song et al., 2020). Although, they were trained in a purely supervised manner on relatively small datasets, the results outperform threshold-based state-of-the-art methods significantly. This corresponds to a recent benchmark on cloud segmentation methods (Hasenbalg et al., 2020). Different threshold-based methods and a CNN were evaluated on a diverse dataset of 829 manually segmented ASIs. In this comparison the CNN performed best.

However, most techniques presented in the literature still focus on binary segmentation and do not differentiate between cloud types. Until today, cloud classification has been mainly studied on image-level, independent of the segmentation approach. Therefore, many datasets contain cutouts of ASIs ("sky patches"). Others are based on camera images with a smaller field of view or ambiguous ASIs were omitted entirely.

For classification, most approaches are learning-based. Various classifiers, like k-nearest neighbors, support vector machines, and CNNs have been trained to recognize the depicted cloud type (Heinle et al., 2010; Zhuo et al., 2014; Ye et al., 2017; Zhang et al., 2018). The classes are usually based on the main cloud genera, sometimes combining visually similar types.

Recently, the combination of both tasks, leading to semantic segmentation has been targeted as well. In two studies, clouds were distinguished in thin and thick clouds (Dev et al., 2015, 2019), however only considering a small dataset of 32 images of sky patches. To our knowledge, there is only one work of an extensive segmentation approach which is based on 9 cloud genera using 600 labeled ASIs (Ye et al., 2019). The authors propose to extract and transform a set of features for generated super-pixels and classify each of them using a SVM. They evaluate their method by comparing the results with a CNN that could not achieve the same accuracy. The problem with deep learning in this case is the lack of data to learn relevant features



and complex data correlations to distinguish between cloud types. Therefore, we propose self-supervised pretraining to enable the model to better learn complex features.

Self-supervised learning is a form of unsupervised learning that does not require manually created labels but generates pseudolabels from the data itself. A model is trained by solving a pretext task, a pre-designed task for learning data representations. Afterwards, representation learning is evaluated in a downstream task. Usually this is done by applying transfer learning, thus using pretrained weights as initialization, and fine-tune a model on a small labeled dataset. In the field of natural language processing, it has become common practice to pretrain a so-called language model, for instance by predicting the following

word in a text (Howard and Ruder, 2018). Lately also in computer vision a trend to self-supervised learning can be observed (Radford et al., 2015; Doersch et al., 2015; Pathak et al., 2016; Noroozi and Favaro, 2016; Lee et al., 2017; Caron et al., 2018).

    In this work we apply two different pretext tasks for self-supervised learning. The first comprises two sub-tasks to be solved. One is to fill cropped areas (Pathak et al., 2016) and the other is to increase the image resolution (Johnson et al., 2016). We refer to it as Inpainting and Superresolution (IP-SR) method. Secondly, we apply the winner of a benchmark on self-supervised

learning for computer vision (Jing and Tian, 2020) which is called DeepCluster (Caron et al., 2018). It is based on an iterative process of clustering the feature outputs from a deep net and using the cluster assignments as pseudolabels for classification.

    By applying self-supervised learning, the limits of a purely supervised approach involving time-consuming and manual creation of ground truth segmentation masks can be overcome. Consequently, the models can be trained with much more data, learning more general and complex features to distinguish cloud types. To our knowledge, this work presents the first

approach of applying deep learning for semantic cloud segmentation on unlabeled data and a new classification of clouds into three layers. The remainder of this work is organized as follows: In sect. 2, the datasets for supervised fine-tuning and self-supervised pretraining are presented. Section 3 introduces the model architecture and the chosen hyperparameters for training. In sect. 4, the trained models are evaluated. First, the results on the pretext tasks are analyzed. Afterwards, the performance of semantic segmentation using self-supervision is compared to a randomly initialized model and another one pretrained on

ImageNet. Then the results on binary segmentation are compared to the results of a CSL on the same dataset. Finally, we conclude our work and provide a brief outlook in sect. 5.

## 2   Cloud image datasets

In this section, the data used for training is described. First, some details about hardware and image properties are given. Then the image selection for the labeled and the unlabeled datasets are discussed.

### 2.1   Image acquisition

All images for training and validating our models were taken at CIEMAT's[1] Plataforma Solar de Almeria (PSA). It is located in the desert of Tabernas (Spain) where atmospheric conditions are often clear, but the observed cloud formations are versatile and multi-layered. For our datasets we used a single all-sky imager based on an off-the-shelf surveillance camera from Mobotix

---

[1]Centro de Investigaciones Energéticas, Medioambientales y Tecnológicas: A spanish research institute with focus on energy and environmental issues.

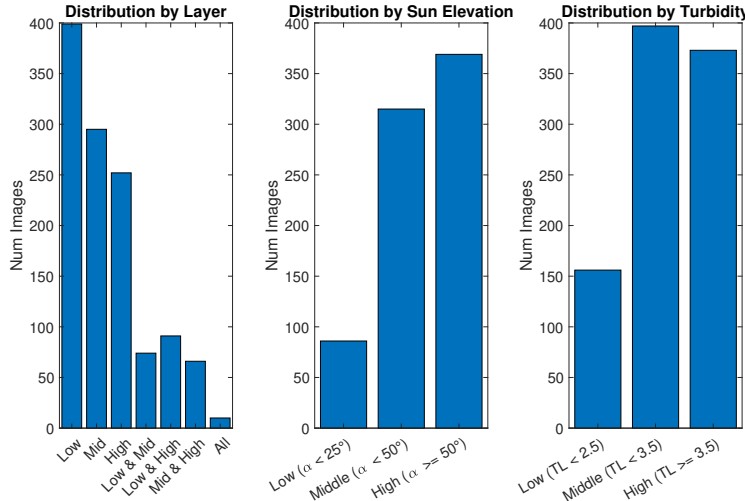

**Figure 1.** Distribution of labeled dataset with respect to cloud types, sun elevation ($\alpha$) and Linke turbidity (TL)

(model Q25). Images are captured and stored with a resolution of 4.35 MP. However, they are cropped and resized to a square

format of 512x512 pixels for the input of the network. Moreover, the images are preprocessed by overlaying a camera mask that removes static objects from the site surroundings. Exposure time is fixed to 160 $\mu$s and no solar occulting devices are installed. The camera is set to take a picture every 30 s from sunrise to sunset, resulting in approximately 1000 to 1600 images per day.

## 2.2    Labeled dataset

For differentiating between cloud types, we categorize clouds into three classes: low-, middle- and high-layer clouds. They combine the 10 main genera defined by WMO (Cohn, 2017) depending on typical cloud base heights. While low-layer clouds are usually dense and heavy clouds of liquid water, high-layer clouds are generally thinner and contain ice particles only. Mid-layer clouds occur in between and contain varying amount of water and ice particles. Consequently, also the optical characteristics of clouds within these layers are different. Another reason for classifying clouds into three layers is to detect

multi-layer conditions. Particularly for solar irradiation forecasting, it is important to determine cloud dynamics which often vary in direction and propagation speed for clouds of different layers.

Our labeled dataset is based on a selection from Hasenbalg et al. (2020). As the original ground-truth segmentation masks used in Hasenbalg et al. (2020) are only binary, we revised 669 images and segmented 101 new images from the same camera, all captured in 2017. In particular images containing thin high-layer clouds and difficult multi-layer conditions were added,

such that a more balanced distribution of cloud types was attained. Furthermore, the selection covers a large variety of sun elevations and Linke turbidity (TL), representing diverse atmospheric conditions. In Fig. 1 an overview of the dataset is given.





### 2.3 Unlabeled dataset for self-supervised pretraining

The unlabeled dataset includes ASIs from the whole year 2017 covering a large variety of conditions. To reduce computation effort, the dataset was filtered, neglecting images that do not contribute much to the learning process. Especially, images with
clear sky conditions are not very useful. Therefore, approximately 40% of all images were sorted out using reference direct normal irradiation (DNI) measurements and a classification procedure as described in Nouri et al. (2019a). As a result, this dataset comprises 286477 ASIs.

### 3 Experimental setup and implementation

In this section details about the model architectures and hyperparameters for supervised segmentation and pretraining are
provided.

### 3.1 Details of the segmentation model

The architecture of our deep learning model is based on a U-Net (Ronneberger et al., 2015). A U-Net is a fully convolutional network (Long et al., 2015) which is composed of an encoder and a decoder part. The encoder part represents the usual downsampling path and consists of a standard ResNet34 (He et al., 2016) in our case. The decoder uses deconvolutions to
upsample the resulting dense representations to the original input size. The special feature of U-Nets is the symmetrical, u-shaped structure of encoder and decoder with skip connections in between. These connections concatenate the output of a directly preceding layer with the the batch normalized output (Ioffe and Szegedy, 2015) of the respective encoder layer. Hence, feature channels in the decoder contain context information which is propagated to higher resolutions enabling precise localization of features. The expansion of the input feature map itself, thus creating new pixels inbetween, is achieved by
applying the so-called pixel-shuffle method with ICNR (Shi et al., 2016; Aitken et al., 2017). Overall, our decoder consists of multiple blocks performing input expansion/merging and applying convolutions/non-linear activations (see Fig. 2). In total there are four of these blocks, and a final output layer producing an output tensor of 5x512x512 (one dimension representing the border part of the ASI). The cloud class ($y$) is then predicted for each pixel ($z$) by applying softmax ($\sigma$) over the 5 channels ($C = 5$) and computing the arg max:

$$\sigma(\boldsymbol{z})_j = \frac{e^{z_j}}{\sum_{k=1}^{C} e^{z_k}} \quad j = 1...C \tag{1}$$

$$y(\boldsymbol{z}) = \arg\max_j \sigma(\boldsymbol{z})_j \tag{2}$$

An overview of the entire architecture is shown in Fig. 3.

For data augmentations we apply 90° rotations and horizontal/vertical flips with a probability of 75% for each batch. In case of the self-supervised models, the input is normalized color-channelwise by subtracting the mean and dividing by the standard
deviation of the unlabeled dataset. We use standard cross entropy as loss function and the Adam optimizer (Kingma and Ba, 2014) with default parameters. Weights of non-pretrained network parts are initialized with Kaiming-init (He et al., 2015) and





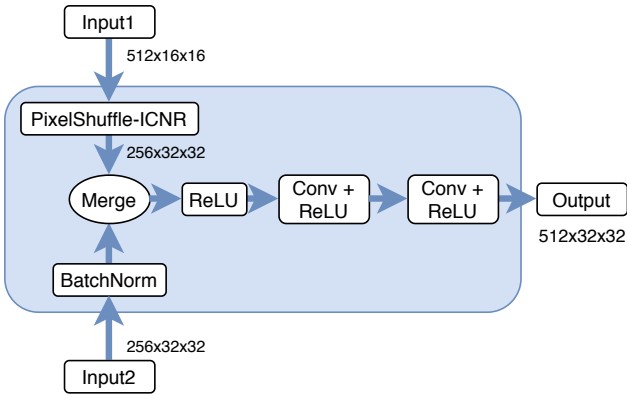

**Figure 2.** Diagram of exemplary inner deconvolution block. Input1 refers to the output from the preceding layer whereas Input2 indicates a skip connection from the corresponding layer in the encoder part. After merging, convolutions and non-linear activations (ReLU) are applied to produce the upsampled output of this block.

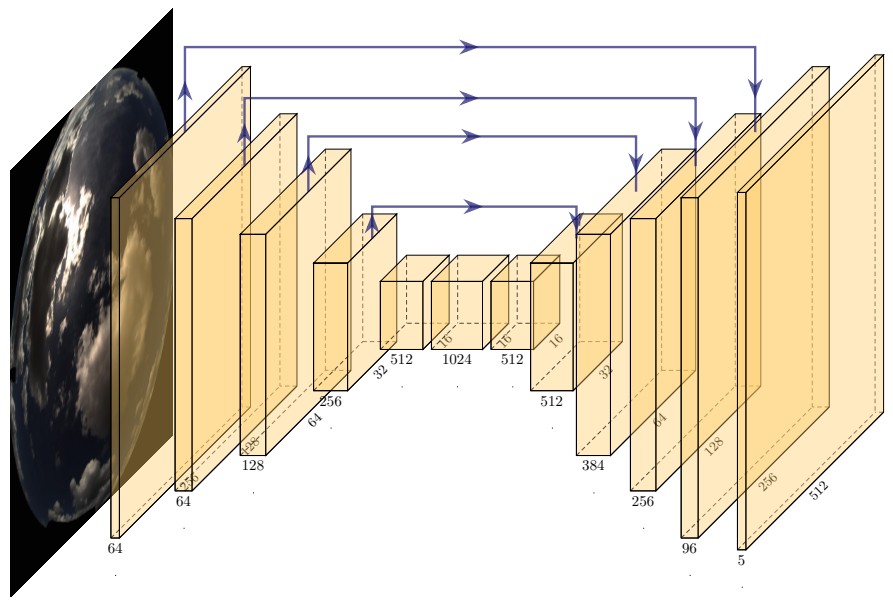

**Figure 3.** Simplified graph of U-Net architecture for segmentation with ResNet34 backbone.

we apply weight decay to prevent overfitting. Furthermore, we apply a learning rate finder and one-cycle policy as described in Smith (2017, 2018). To obtain faster convergence, we split training into two phases. In the first phase, the pretrained encoder part is frozen for 20 epochs using a larger learning rate. Afterwards, the entire network is fine-tuned with a smaller one for another 20 epochs. These values were chosen after examining the loss curves for training and validation set for longer training runs. The network is trained in batches of 4 images using 80% of the dataset which leaves 154 images for validation. All





**Table 1.** Hyperparameters for training the segmentation model.

| | |
|---|---|
| Input size | 512x512 |
| Normalization mean | [0.1739, 0.1696, 0.1715] |
| Normalization std | [0.1376, 0.1297, 0.1175] |
| Learning rate frozen | 1e-3 |
| Learning rate unfrozen | 1e-4 |
| Weight decay | 1e-2 |
| Fixed validation | 20% |
| Batch size | 4 |
| Num epochs frozen | 20 |
| Num epochs unfrozen | 20 |

models are trained on a single GPU (Nvidia GeForce RTX 2080 Super) and were implemented using fastai v1, a high-level Pytorch API. A summary of the hyperparameter selection is given in Table 1.

### 3.2 Pretext tasks for self-supervised learning

As mentioned before, we implemented two methods for self-supervised learning based on techniques which proved successful in the literature: IP-SR and DeepCluster.

For the **IP-SR** task, the original image is corrupted by inserting four black squares and by reducing the resolution by half. The square boxes are positioned randomly within the ASI and have an edge length of 80 pixels. This size was chosen such that smaller clouds can be occluded, but general cloudiness conditions can still be determined by human observers. Furthermore,

after downsizing the images they need to be upscaled again to match the network's input size which is achieved using bilinear interpolation. For inpainting, the model should learn to predict the missing parts by observing and recognizing the surrounding conditions. Regarding superresolution, the model should learn structural and textural characteristics of clouds which helps to better distinguish cloud types in later segmentation. For this task, the architecture is the same as for the segmentation model (apart from the output layer), and also the hyperparameters are mostly equal. The loss is composed of a pixel-wise MAE and

a so-called perceptual loss (Johnson et al., 2016). Due to limited GPU memory and high computational efforts, batch size was limited to 2 and training consists of 10 epochs.

The **DeepCluster** method (Caron et al., 2018) follows the approach of creating pseudo-labels that can be used for classification. Thereby, the network should learn useful data representations that are relevant to recognize objects or in our case clouds. It consists of two alternating steps. (1) The output features of the CNN (ResNet34 encoder part) are assigned to a pre-

defined number of clusters. For this we apply a standard k-means algorithm. (2) The resulting cluster assignments can then be used as pseudo-labels to solve a standard classification problem. After each epoch (clustering + classification), the features are



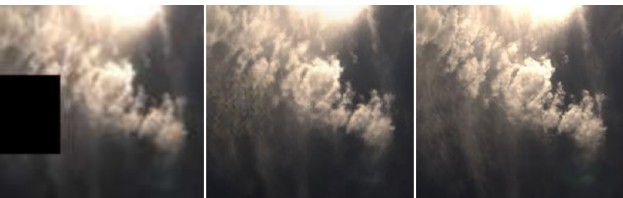

**Figure 4.** From left to right: input, prediction and ground truth in inpainting-superresolution pretext task.

clustered again leading to new pseudo-labels. According to Caron et al. (2018), k should be set higher, than the target classes. Hence, to choose a reasonable value for k, we evaluated three models using $k = \{30, 100, 1000\}$. Hyperparameters were mostly adopted from the original DeepCluster setup, only batch size and the number of epochs were set to 32 and 50 respectively.

For both pretext tasks (IP-SR and DC), we tested two weight initializations. First, standard random (Kaiming) init, i.e. the network starts with random weights. Secondly, initialization with pretrained ImageNet weights. Hence, the ResNet34 part is initialized with weights of a ResNet34 model that was trained on ImageNet. These are available online and can be downloaded via Pytorch.

## 4 Experimental results

Next, we briefly examine the models from self-supervised pretraining. However, the major part deals with the segmentation results on the validation set of our labeled data.

### 4.1 Pretraining results

The pretraining results are only evaluated qualitatively to check whether our models learned to solve the tasks sensibly. Regarding **IP-SR**, Fig. 4 shows an ASI section of the input, the prediction and the ground-truth (original) image. It can be seen

that the part of the black box is indeed filled similarly to the Cirrus clouds which were occluded. Even small parts of the Altocumulus clouds in the upper-right corner are reconstructed. Also more structural and textural details are present in the predicted image compared to the input. However, there are also some artifacts visible in the reconstructed area and there is still a notable difference to the original resolution. Nevertheless, the model is able to reconstruct the original images similarly in all inspected cases so that we can assume a successful learning effect.

In the case of **DeepCluster**, we examined exemplary samples that were classified based on the final cluster assignments. Figure 5 depicts four randomly picked ASIs of four different clusters. In this example, the model was trained with $k = 30$ clusters. It can be observed that the clusters consider general cloudiness conditions like cloud coverage but also focus on sun elevation and turbidity. In the upper row, even raindrops on the lenses seem to be a characteristic feature for this cluster assignment.





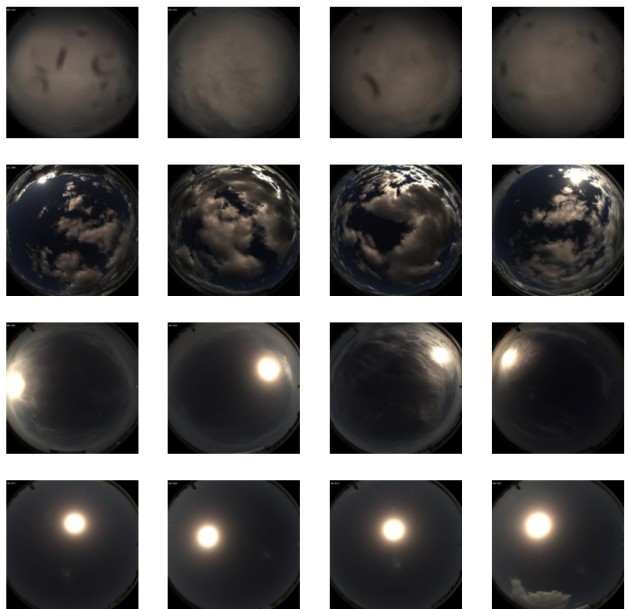

**Figure 5.** Four Randomly chosen samples of four clusters after training. The images of each row correspond to one cluster.

## 4.2 Segmentation results

To evaluate overall segmentation performance, we use two commonly applied metrics. First, pixel-accuracy which is defined by the number of correctly predicted ASI pixels divided by the number of all ASI pixels ($NumPix$). Secondly, mean intersection over union (mean IoU): the overlapping area of predicted and target pixels by their union. The border part of the ASI, indicating masked image areas (see sec. 2.2), is neglected as this would distort the results. Furthermore, we also evaluate precision, recall and IoU for each class, to analyze segmentation in more detail. These metrics are normalized corresponding to the number of (predicted/target) pixels of the respective class. Thus, the size of an observed cloud type is considered for computing the metrics' averages. Our evaluation metrics are therefore defined as:





**Table 2.** Comparison of segmentation results when testing different training setups for self-supervised pretraining.

| | IP-SR | | DeepCluster | | | | | |
|---|---|---|---|---|---|---|---|---|
| Pretraining init | Random | ImageNet | Random | | | ImageNet | | |
| Num clusters k | - | - | 30 | 100 | 1000 | 30 | 100 | 1000 |
| Pixel-accuracy | 85.09 | **85.75** | 84.40 | 84.46 | 84.60 | 85.22 | 84.86 | 84.88 |
| Mean IoU | **80.58** | 80.46 | 79.81 | 79.58 | 80.09 | 80.48 | 80.41 | 80.22 |

$$pixelAcc = \frac{1}{N} \sum_{i=1}^{N} \frac{TP_i + TN_i}{NumPix} \tag{3}$$

$$mIoU = \frac{1}{N} \sum_{i=1}^{N} \frac{TP_i}{TP_i + FP_i + FN_i} \tag{4}$$

$$precision_c = \frac{\sum_i^N TP_{c,i}}{\sum_i^N TP_{c,i} + FP_{c,i}} \tag{5}$$

$$recall_c = \frac{\sum_i^N TP_{c,i}}{\sum_i^N TP_{c,i} + FN_{c,i}} \tag{6}$$

$$IoU_c = \frac{\sum_i^N TP_{c,i}}{\sum_i^N TP_{c,i} + FP_{c,i} + FN_{c,i}} \tag{7}$$

$$\tag{8}$$

where $TP, FP$ $(TN, FN)$ refers to true/false positives (negatives), the index $c$ to one of the cloud classes and $N$ to the number of ASIs.

First we compared different training setups of the pretext tasks. Apart from pretraining initialization, this comprises also the different values for k in the DeepCluster pretraining. After pretraining, the trained weights of the ResNet34 part are transferred to the segmentation model, fine-tuned on the training set and evaluated on the validation set. Table 2 summarizes the results for pixel-accuracy and mean IoU. In all setups the differences are rather small. Pixel-accuracy is between 84.6% and 85.8% whereas mean IoU is always slightly over 80%. Hence, no significant improvement using ImageNet initialization for self-supervised pretraining can be observed. The influence of $k$ on the segmentation task is also small, but smaller values seem to lead to marginally better results.

In the next step, we compare our self-supervised approach to standard supervised training with random and ImageNet initialization. All following statements are based on Table 3. For a better overview, only the best self-supervised models in terms of pixel-accuracy are considered here (DC and IP-SR with ImageNet init and DC with $k = 30$). Regarding overall pixel-accuracy and mean IoU, our self-supervised approaches reach over 3% points more than starting with ImageNet and about 7% points more than with random initialization. Concerning the prediction of cloud classes, a more significant improvement





**Table 3.** Segmentation results comparing our self-supervised approaches with standard ImageNet or random initialization. IP-SR* and DC**
refer to pretraining starting with ImageNet weights and $k = 30$ for the DC method.

|  | Class | Random | ImageNet | IP-SR* | DC** |
|---|---|---|---|---|---|
| Pixel-accuracy | - | 78.34 | 82.05 | **85.75** | 85.22 |
| Mean IoU | - | 72.11 | 77.10 | 80.46 | **80.48** |
| Precision | Sky | 90.92 | 93.67 | 94.06 | **94.25** |
|  | Low-layer | 63.61 | 69.70 | **78.75** | 73.70 |
|  | Mid-layer | 49.14 | 56.34 | 71.23 | **75.09** |
|  | High-layer | 48.72 | 58.67 | 64.41 | **64.73** |
| Recall | Sky | **97.95** | 97.53 | 97.37 | 97.13 |
|  | Low-layer | 71.43 | 78.74 | 85.58 | **89.13** |
|  | Mid-layer | 26.21 | 32.38 | **48.88** | 40.15 |
|  | High-layer | 47.32 | 65.06 | 68.76 | **70.65** |
| IoU | Sky | 89.23 | 91.50 | **91.73** | 91.69 |
|  | Low-layer | 50.71 | 58.66 | **69.52** | 67.63 |
|  | Mid-layer | 20.62 | 25.88 | **40.82** | 35.43 |
|  | High-layer | 31.59 | 44.61 | 49.83 | **51.02** |

becomes evident. On average, precision, recall and IoU are about 9-10% points higher for the self-supervised IP-SR approach
compared to ImageNet init. For the mid-layer class, it is about 15% points. Also our DC method outperforms the purely

supervised approaches significantly, reaching similar values on average as the IP-SR method. Overall, self-supervised learning
achieves the best results except for recall of *sky*. However, this is negligible as all approaches reach values over 97% and
corresponding precision is significantly lower, indicating overestimation.

To further interpret segmentation results, we analyze misclassification using a confusion matrix shown in Fig. 6. Most
frequent confusions are between adjacent cloud layers which is why the mid-layer class is predicted worst. Also high-layer

clouds are sometimes not detected at all. On the other hand, low- and high-layer clouds can be distinguished quite reliably.

When examining exemplary segmentation masks, another problem becomes apparent, see Fig. 7. Thinner parts of low-layer
clouds are often misclassified as mid- or even high-layer clouds, since they typically occur in twilight zones. Moreover, the
decrease in classification accuracy for lower elevation angles is expected, because the fish-eye lenses capture these areas with
lower resolution. Another challenge are stratus-like overcasts. These clouds often lack texture, they can have variable depth

and they can occur in all layers which makes it particularly hard to differentiate.



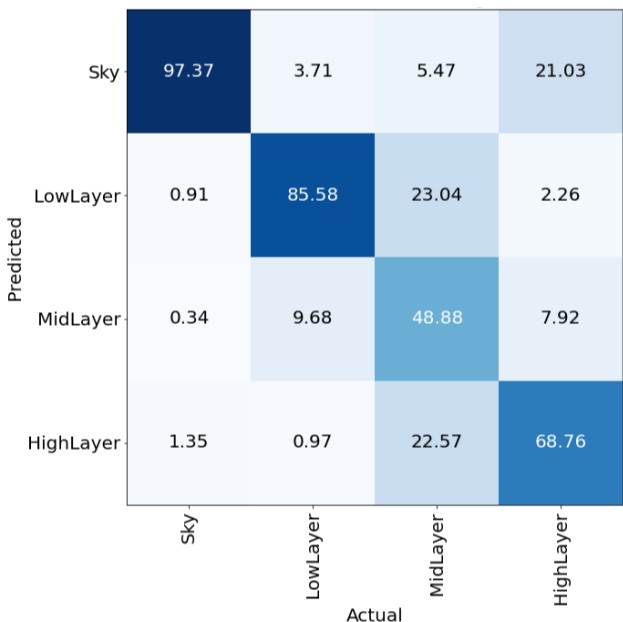

**Figure 6.** Confusion matrix of segmentation model with IP-SR* pretraining.

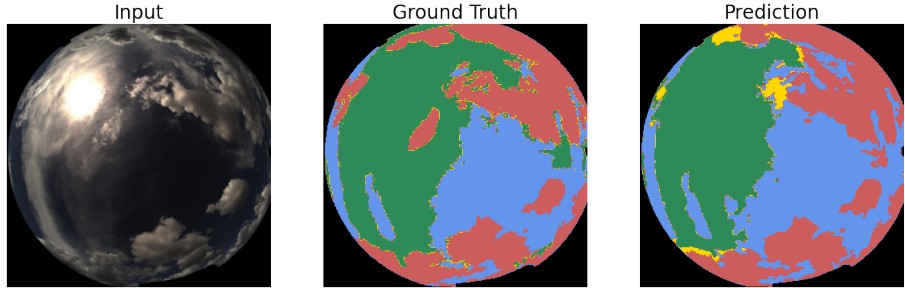

**Figure 7.** Exemplary predicted segmentation mask compared to ground truth for a given input from the validation set. (blue: sky, red: low-layer, yellow: mid-layer, green: high-layer).

## 4.3 Comparison of binary segmentation results

Finally, we compare our approach on binary segmentation with the results of a state-of-the-art CSL (Kuhn et al., 2018). The dataset is the same as for previous evaluations, comprising 154 representative ASIs. Beforehand, our segmentation model (IP-SR*) was fine-tuned on binary ground-truth masks, leading to slightly better results than postprocessing the 4-class masks. Again we evaluated the models on pixel-accuracy. Reaching 95.2% on average, the CNN outperforms the CSL (87.9%) by over 7%-points. As shown in Fig. 8, we also analyzed both methods under different predefined conditions of cloudiness where the





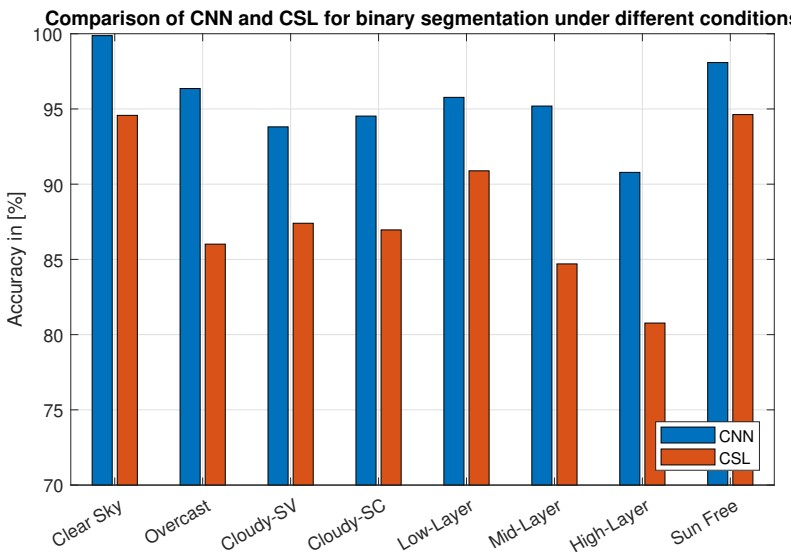

**Figure 8.** Comparison of our results with a CSL on the validation set under different conditions of cloudiness. Cloudy-SV and Cloudy-SC specify partially clouded conditions with the sun visible (SV) or covered (SC). Sun Free determines whether the sun disk is completely free from clouds.

CNN always outmatches the CSL. Especially for more challenging mid- and high-layers clouds, our model achieves 90-95%. In comparison to the CSL this a benefit of over 10%-points.

The binary cloud detection is a processing step at the beginning of most ASI applications, such as nowcasting. Hence, the
increase in accuracy has the potential to improve the overall performance of these ASI applications significantly.

### 4.4  Comparing our results to the literature

In principle, an expressive comparison to the literature can only be conducted if the data for evaluation is the same. That is because prevailing cloud conditions within the dataset can be very different and thus better or worse accuracies can be achieved. For example, the amount of ASIs containing difficult Cirrus clouds or atmospheric conditions with high Linke
turbidity can affect the overall accuracies significantly. Still, latest developments towards learning-based approaches show the superiority over traditional threshold-based methods with our results confirming this trend (see Table 4). For instance, the proposed learning-based model from Cheng and Lin (2017) achieves over 10% points more than a standard red-blue ratio, or the HYTA model (Li et al., 2011). Also in other recent studies, machine learning models clearly outperform classical approaches like red-blue ratio, CSLs or the HYTA model.





**Table 4.** Comparison of our results with the literature. Number of ASIs refers to the number of images used for validation.

| Source | Num classes | Num ASIs | Method | Acc Semantic | Acc Binary |
|---|---|---|---|---|---|
| Cheng and Lin (2017) | 2 | 250 | Red-blue ratio | - | $\sim 78\%$ |
| | | | HYTA | - | $\sim 79\%$ |
| | | | Feature Extraction + Classifier | - | $\sim 90\%$ |
| Ye et al. (2019) | 9 | 460 | Feature Extract./Transform. + Classifier | 71.28% | 93.71% |
| Hasenbalg et al. (2020) | 2 | 160 | CSL | - | 92.51% |
| | | | HYTA+ | - | 95.79% |
| | | | CNN | - | 96.98% |
| Xie et al. (2020) | 2 | 60 | Red-blue ratio | - | 81.17% |
| | | | CNN | - | 96.24% |
| This work | 2 | 154 | CSL | - | 87.88% |
| | 4 | 154 | CNN | 85.75% | 95.15% |

## 275 5 Conclusions

In this paper, we presented the first approach to pretrain deep neural networks for ground-based sky observation using raw image data. Based on our results, these networks can be trained more effectively without the need of labeling thousands of images by hand. For cloud segmentation, this offers the possibility to simultaneously detect and distinguish clouds with associated properties in ASIs using deep learning. Our developed segmentation model is based on the U-Net architecture with

a ResNet34 encoder and the model categorizes ASI pixels into four classes. For self-supervised pretraining, we evaluated two distinct pretext tasks: Inpainting-Superresolution and DeepCluster. We inspected the pretrained models on solving their respective tasks and evaluated them on our validation set for cloud segmentation, containing 154 ASIs. By comparing the results from our pretrained models with the ones from ImageNet or random initialization, we showed the benefits of self-supervised learning. Considering only pixel-accuracy, our pretrained models reach over 85%, compared to 82.1% (ImageNet init) and

78.3% (random init). For mean IoU, the results are 80.5%, 77.1% and 72.1% respectively. But the most significant advantage becomes evident when regarding the distinction of cloud classes. In particular for more challenging cloud types such as mid- or high-layer clouds, precision, recall and IoU of the self-supervised models are often 10-20% points higher. Furthermore, we evaluated our approach on binary segmentation. Compared to a state-of-the-art CSL our model is more accurate under all examined conditions. On average accuracy is 95.2% and thus 7% higher than the accuracy of the CSL. Although an expressive

comparison to the literature due to different datasets is not possible, our work confirms the superiority of learning-based approaches for cloud segmentation. However, there is still room for improvement in recognizing cloud types. High variability and similarity of different types makes precise cloud classification difficult. In particular distant clouds and twilight zones cause false cloud type predictions. Therefore, more studies on other pretext tasks or other methods exploiting raw image data



are needed. Finally, as self-supervised pretraining uses much more image data for training, it can be expected that the resulting
models are more robust when being applied on other datasets. In particular, future models could be trained on large datasets of
multiple cameras at different sites, potentially capable of generalizing well on any camera.

*Author contributions.*  BN, SW, NB, YF conceived the presented idea. YF designed the models, performed the computations and analyzed
the data. BN, SW, NB, RT verfified the analytical methods and supervised the findings of this work. RT encouraged YF to investigate
unsupervised learning methods. YF, MH contributed to the preparation of the dataset. PK, MH developed the reference Clear-Sky Library.
YF prepared the manuscript with contributions from all co-authors. RP supervised the project.

*Competing interests.*  The authors declare that they have no conflict of interest.

*Acknowledgements.*  The German Federal Ministry for Economic Affairs and Energy funded this research within the WobaS-A project (Grant
Agreement no. 0324307A). Further funding was received by the European Union within the H2020 program under the Grant Agreement no.
864337 (Smart4RES).





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
