# Peer review of "Applying self-supervised learning for semantic cloud segmentation of all-sky images"

_Atmospheric Measurement Techniques, 2021_

## Author Response (AR1)

**Author's response document for research article "Applying self-supervised learning for semantic cloud segmentation of all-sky images"**

**Referee Comment 1**

This manuscript describes a new method for classifying All Sky Images. This matter, the analysis of sky images in order to recognize different cloud types is of interest in atmospheric and climate research, and also for applications such as solar power plant management. The main novelty of the method is the self-supervised techniques used in the development. In my opinion, the paper is well written and correctly structured, although I cannot give a well-founded criticism on the methodological section, given my limited knowledge of artificial intelligence techniques.

I only have a number of minor points to comment, which may eventually lead to introduce some little changes in the final version of the paper:

**Reply on RC1**

Thank you very much for your detailed review and the provided feedback. We answer to your comments below after each comment.

Line 24. Clouds play a significant role in the climate system and therefore in climate change. I wouldn't write "in global warming".

- Thank you. We replaced global warming with climate change.

Line 34. Use "complement" instead of "addition".

- Changes as suggested.

Figure 1. I'm not exactly sure about what is represented in this figure. The total number of images seems different in the three panels: in an approximation, bars totalize 1200 images in the first panel, 770 in the second, and 910 in the third. If it is the same dataset, the number of images should be the same, shouldn't it? Last paragraph in page 5 refers also to the number of images. According with this, the dataset contains 770 images. So you may need to explain a little more clearly how Fig. 1 in created. Moreover, could you also explain how did you estimate Linke turbidity?

- Thank you for spotting this issue. Figure 1 and the caption were updated, as different sets were shown in the turbidity and sun elevation subplots. The individual plots in the figure now refer to the same dataset of 770 images. However, the first plot from the left does not

represent mutually exclusive labels but counts each ASI containing the respective cloud layer (or combination of cloud layers) independently. For instance, an ASI with low- and high-layer clouds counts for low-layer, high-layer as well as low-layer & high-layer. This shows the reader how many ASI contain which cloud layer and how many images are combinations of multiple cloud layers. A remark was also added to the figure's caption.

Furthermore, for the computation of the turbidity coefficient a reference to Ineichen, P. and Perez, R (2002) was added

Line 158-159. Can you explain further what do you mean by "one dimension representing the border part of the ASI"?

- Regarding Lines 158-159, we clarified what we meant with "border" part of the ASI. An additional dimension of the output tensor is required to account for the black border/periphery of the ASI produced by the fish-eye camera. As the model is trained with square images (512x512), parts of these black pixels of the periphery are contained in the input tensor and need to be distinguished from the actual ASI. Thus, the output tensor (also of size 512x512) includes this "dummy" dimension which does not represent a cloud type or clear sky, but it represents the periphery of the circular ASI. It can also be seen as an extra class for the background that is needed since each pixel will be assigned a class.

First paragraph in Section 4.1. You could emphasize that this is only one example. How general are the comments that you make for this example?

- We updated section 4.1 to emphasize that it is only an example and that we cannot conclude the general performance from this example alone. However, we also added a note, that we visually inspected images from several days and could not find any unexpected predictions. We replaced the text:

"Regarding **IP-SR**, Fig. 4 shows an ASI section of the input, the prediction and the ground-truth (original) image. It can be seen that the part of the black box is indeed filled similarly to the Cirrus clouds which were occluded. Even small parts of the Altocumulus clouds in the upper-right corner are reconstructed. Nevertheless, the model is able to reconstruct the original images similarly in all inspected cases so that we can assume a successful learning effect."

by

"Regarding **IP-SR**, Fig. 4 shows an example of an ASI section of the input, the prediction and the ground-truth (original) image. Here, it can be seen that the part of the black box is indeed filled similarly to the Cirrus clouds which were occluded. Even small parts of the neighboring Altocumulus clouds in the upper-right corner are reconstructed. Also, more structural and textural details are present in the predicted image compared to the input. However, there are also some artifacts visible in the reconstructed area and there is still a notable difference to the original resolution. We chose this particular image section, as the occluded part contains two cloud layers and it can be seen that the model's predictions for the black box depend on the surrounding conditions. Clearly, this exemplary instance does not prove the general capability of the model to make reasonable predictions for the entire dataset. However, as we could not find any unexpected pixel reconstructions on multiple days of generated ASIs under various conditions, a successful learning effect can be assumed."

Figure 5. Are images really random? Could you add the date and time in each image, in order for the reader to see that they do not correspond to nearly consecutive images (or images taken the same day under similar conditions)?

- To show that the images from Figure 5 are random, we included a timestamp to the individual images. Apart from two images they are also all from different days.

**Referee Comment 2**

Manuscript „Applying self-supervised learning for semantic cloud segmentation of all-sky images" by Yann Fabel and coauthors describes application of new developments in neural networks to improve automatic cloud recognition in all-sky images. In order to increase the size of training data set used in supervised learning of the deep neural network, they apply a two-step learning procedure. In the first step, „self-supervised" pretraining of the network, a vast volume of 300000 images is used in two distinctive approaches based on image reconstruction and cluster analysis. In the second, fine tuning stem, about 600 images is used for training and ~150 for verification.

As the effect of the procedure, the authors report significant improvement of the recognition compared to existing automated procedures, reaching 95% pixel accuracy.

The manuscript is clearly written, providing step-by-step description of the activities. All the necessary information is given.

The discussion part, however, could be improved. While in Fig. 6 confusion matrix is presented and in Fig. 7 exemplary segmentation mask from one of the recognition is presented, it would be useful to read/see some examples of best and worst recognition to compare. The authors describe which regions and types of clouds are most problematic, yet it would be informative to show examples.

I believe that such minor revision of the document would give useful information and possibility co compare whether other recognition algorithms have similar for different problems.

After such a minor revision the manuscript van be accepted for final publication.

**Reply on RC2**

Thank you very much for your revision and the feedback you provided. It is indeed helpful for the reader to also see some good and bad examples of the model's performance instead of reading the plain numbers only. Therefore we added a bad example where the model was able to differentiate between cloud and sky, but could not recognize the individual cloud layers very accurately. The chosen ASI depicts difficult conditions with overlapping cloud layers under turbid conditions which do not occur very frequently and which probably led to these rather bad prediction.

We replaced these short two sentences

When examining exemplary segmentation masks, another problem becomes apparent, see Fig. 7. Thinner parts of low-layer clouds are often misclassified as mid- or even high-layer clouds, since they typically occur in twilight zones.

by the a more comprehensive text and added Fig. 7b:

When examining exemplary segmentation masks as depicted in Fig. 7, another problem becomes apparent. Thinner parts of low-layer clouds are sometimes misclassified as mid- or even high-layer clouds, since they typically occur in twilight zones. Moreover, the decrease in classification accuracy for lower elevation angles is expected, because the fish-eye lenses capture these areas with lower resolution. Another challenge are stratus-like overcasts. These clouds often lack texture, they can have variable depth and they can occur in all layers which makes it particularly hard to differentiate. A very challenging cloud condition is shown in Fig. 7b, which combines a lot of the challenges just mentioned. Although it recognizes parts of the low-layer clouds, the model seems to be uncertain about the specific cloud layers, as it often changes between all three classes. Nevertheless, the accuracy regarding cloudy and cloud-free pixels is still high.

[Figure]

(a) Example of accurate cloud layer prediction

(b) Example of inaccurate cloud layer prediction

Figure 7. Exemplary predicted segmentation mask compared to ground truth for a given input from the validation set. (blue: sky, red: low-layer, yellow: mid-layer, green: high-layer)